# Glutamine Starvation Affects Cell Cycle, Oxidative Homeostasis and Metabolism in Colorectal Cancer Cells

**DOI:** 10.3390/antiox12030683

**Published:** 2023-03-10

**Authors:** Martina Spada, Cristina Piras, Giulia Diana, Vera Piera Leoni, Daniela Virginia Frau, Gabriele Serreli, Gabriella Simbula, Roberto Loi, Antonio Noto, Federica Murgia, Paola Caria, Luigi Atzori

**Affiliations:** Department of Biomedical Sciences, University of Cagliari, Cittadella Universitaria, SS 554, km 4.5, 09042 Monserrato, Italy

**Keywords:** colorectal cancer, cancer cell metabolism, glutamine starvation, energetic pathways, antioxidant defenses, metabolomics

## Abstract

Cancer cells adjust their metabolism to meet energy demands. In particular, glutamine addiction represents a distinctive feature of several types of tumors, including colorectal cancer. In this study, four colorectal cancer cell lines (Caco-2, HCT116, HT29 and SW480) were cultured with or without glutamine. The growth and proliferation rate, colony-forming capacity, apoptosis, cell cycle, redox homeostasis and metabolomic analysis were evaluated by 3-(4,5-dimethylthiazol-2-yl)-2,5-diphenyltetrazolium bromide test (MTT), flow cytometry, high-performance liquid chromatography and gas chromatography/mass spectrometry techniques. The results show that glutamine represents an important metabolite for cell growth and that its deprivation reduces the proliferation of colorectal cancer cells. Glutamine depletion induces cell death and cell cycle arrest in the GO/G1 phase by modulating energy metabolism, the amino acid content and antioxidant defenses. Moreover, the combined glutamine starvation with the glycolysis inhibitor 2-deoxy-D-glucose exerted a stronger cytotoxic effect. This study offers a strong rationale for targeting glutamine metabolism alone or in combination with glucose metabolism to achieve a therapeutic benefit in the treatment of colon cancer.

## 1. Introduction

Colorectal cancer (CRC), with 1.9 million new cases and 935,000 deaths in 2020, ranks third in incidence and second for mortality in both sexes worldwide [1]. Gene alterations [2] and some diseases, such as chronic inflammatory bowel disease [3] and type 2 diabetes [4], are predisposed to the development of CRC. In addition, some lifestyle habits have been recognized as risk factors for the onset of the disease, such as excess body weight [5], cigarette smoking [6], a sedentary lifestyle [7], the consumption of processed meats [8] and a low intake of dietary fiber [9,10]. The incidence in young adults under 50 has increased alarmingly in recent years, as reported by numerous studies [11,12,13,14]. The CRC stage notably influences survival at diagnosis: the 5-year survival rate ranges from 90% if diagnosed in the early stages to 14% if the cancer is already metastatic [14]. Therefore, a better understanding of the etiological mechanisms in order to improve prevention and early diagnosis has become necessary. The high proliferation rate of cancer cells leads to an increase in the demand for energy and bio-macromolecules. Most cancer cells rewire their metabolism to meet this increased requirement, and the reprogramming of energy metabolism has been recognized as a new hallmark of cancer [15]. Notably, several cancer cells become dependent on specific nutrients, including glutamine, and they exhibit glutamine addiction [16,17,18,19]. Glutamine is defined as a “conditionally” essential amino acid. Indeed, although it can be synthesized de novo by cells, it becomes essential in particular pathological conditions and highly proliferating cells, such as cancer cells [20]. Glutamine plays a pivotal role in numerous processes. In particular, it is important for mitochondrial metabolism as it fuels the tricarboxylic acid cycle (TCA) to produce energy [21,22,23]. Recently, glutamine addition has attracted attention as a therapeutic target to selectively affect cancer that depends on the availability of this amino acid to survive and grow, including colorectal cancer [19,24]. However, the overall metabolic deregulation underlying the effects induced by glutamine deprivation on colorectal cancer cells has not yet been clarified. Indeed, metabolomics could represent a promising approach for investigating the alterations implemented by the tumor to survive and grow, with the main goal of identifying early diagnostic biomarkers and therapeutic targets [25,26]. To contribute to this issue, the present work aimed to identify metabolic perturbations due to glutamine deprivation in colorectal cancer cells, exploring which metabolic pathways were modified and which strategies cancer cells put in place to overcome the glutamine deprivation. Moreover, the consequences on survival, proliferation and redox homeostasis caused by glutamine deprivation were highlighted.

## 2. Materials and Methods

### 2.1. Cell Culture

Caco-2 cells were kindly gifted by Prof. Monica Deiana (University of Cagliari, Cagliari, Italy). HCT116 cells were given by Dr. Giuseppina Sanna (University of Cagliari, Italy). HT29 cells were purchased from Elabscience^®^, Houston, TX, USA. SW480 cells were obtained from the cell bank ICLC (San Martino Polyclinic Hospital, Genova, Italy). The genetic background and the tumor of origin classification are reported in detail in Table 1. The cell lines were cultured in Dulbecco’s Modified Eagle’s Medium (DMEM) high glucose supplemented with 10% heat-inactivated bovine serum (FBS, Life Technologies, Milan, Italy), 100 U/mL penicillin, 100 mg/mL streptomycin (Sigma-Aldrich, Milan, Italy), L-glutamine 4 mM (Euroclone, Milan, Italy) and sodium pyruvate 1 mM (Euroclone, Milan, Italy), and were maintained at 37 °C in a humidified 5% CO_2_ atmosphere. Before being deprived of glutamine, the cells were grown for 4 days in a complete medium to reach an adequate number of cells in order to optimally perform all assays.

### 2.2. Growth Curves

The growth curves of CRC cells were obtained by performing MTT ((3-(4,5-dimethylthiazol-2-yl)-2,5-diphenyltetrazolium bromide) tetrazolium) assay [32,33], which allows for identifying viable and metabolically active cells. Briefly, cells were seeded in 96-well plates (1 × 10^5^ cells/mL; 100 μL/well) in a complete medium to allow for attachment. Subsequently, cells were deprived of glutamine. MTT assays were performed every day for a total of 4 days. Firstly, the medium was removed, and cells were rinsed twice with phosphate-buffered saline pH 7.4 (PBS, Euroclone, Pero, Italy). Thereafter, 50 µL of MTT solution (2 mg/mL in PBS) was added and left for 4 h at 37 °C. After that, the PBS was aspirated, and the derived blue-violet formazan was solubilized with 100 µL of DMSO (Sigma-Aldrich, Milan, Italy). The absorbance was measured at 570 nm, after 15 s of shaking, using a microplate reader (Infinite 200, Tecan, Salzburg, Austria). Data were expressed as absorbance at 570 nm ± standard deviation. All experiments were carried out in at least triplicate and repeated 3 times. 

### 2.3. Colony Forming Assay

The proliferation capacity of CRC cells was evaluated through colony forming assay as described by Liang and colleagues [34]. Briefly, cells were seeded at low concentrations in 24-well plates, (1–2 × 10^3^ cells/mL, 500 µL per well) in a complete medium and left to grow for 4 days. Then, cells were deprived of glutamine for a total of 6–10 days depending on cell lines. At this time, cells were fixed with 200 µL of ice-cold methanol for 20 min, rinsed with water and finally stained with a solution of 5% crystal violet in 80% methanol. After 5 min, cells were washed and air-dried overnight. Samples were solubilized with glacial acetic acid, and a microplate reader (Infinite 200, Tecan, Salzburg, Austria) was used to measure optical density at 570 nm, as reported by Wang and colleagues [35]. All experiments were carried out in at least triplicate and repeated 3 times. 

### 2.4. Fluorescence-Activated Cell Sorting (FACS) Analysis

To investigate cell death, a flow cytometric analysis was performed. We used the cell apoptosis kit Annexin V/Propidium Iodide (PI) double staining uptake (Life Technologies, Monza, Italy). CRC cell lines (Caco-2, HCT116, HT29 and SW480) were seeded at 5 × 10^4^ cells/mL in 6-well plates and cultured in a complete medium for 4 days. Subsequently, cancer cells were exposed to a complete medium or glutamine-deprived medium for additional 48 h. Next, they were washed with PBS and 100  μL of annexin binding buffer plus 5  μL of Annexin V and 1 μL of PI were added. After that, cells were incubated in the dark for 15 min at room temperature. Stained cells underwent flow cytometric analysis by measuring the fluorescence emission at 530 and 620 nm using a 488 nm excitation laser (MoFloAstrios EQ, Beckman Coulter, Brea, CA, USA) [33]. The evaluation of apoptosis was performed using Software Summit Version 6.3.1.1, Beckman Coulter, Brea, CA, USA. To explore the effect of glutamine starvation on the cell cycle, a flow cytometric analysis was performed using FxCycle™ PI/RNase Staining Solution kit (Life Technologies). The cells were seeded and left to grow for 4 days in 6-well plates (5 × 10^4^ cells/mL) in a complete medium. Then, they were deprived of glutamine for 48 h. After trypsinization, the cells were washed with PBS and fixed in ethanol for 30 min. After that, the cells were collected and centrifuged. Pellet was washed with PBS, resuspended in a buffer containing PI and then incubated for 30 min. The DNA content was then detected using flow cytometry (MoFloAstrios EQ, Beckman Coulter, Brea, CA, USA) [36]. The analysis of cell cycle phase distribution was performed with Kaluza Flow Cytometry Analysis Software (Software Version 1.2, Beckman Coulter, Brea, CA, USA) by setting 3 gates in each single parameter histogram: G0/G1, S and G2/M. 

### 2.5. Sample Preparation for Metabolomics Analysis

To evaluate metabolomics changes induced by glutamine starvation, intracellular metabolites of Caco-2, HCT116, HT29 and SW480 cells were extracted and analyzed using gas chromatography–mass spectrometry, as described by Santoru and colleagues [37]. Briefly, cells were seeded and left to grow for 4 days in 6-well plates (5 × 10^4^ cells/mL) in a complete medium. Subsequently, cells were starved of glutamine for 48 h. After washing the cells twice with physiological solution, 500 µL of ice-cold 80% methanol solution was used to extract hydrophilic metabolites. The extraction was carried out for 10 min on ice to limit the metabolic reactions and preserve the intracellular compounds. Afterward, cells were scraped and transferred in tubes. To ensure complete lysis of cell membranes, samples underwent 10 min of ultrasonication at 4 °C. Cell suspensions were centrifuged at 4500× *g* for 30 min at 4 °C. Thereafter, 400 μL of supernatant was aliquoted and dried overnight in an EppendorfTM Concentrator Plus. Next, 50 μL of a solution of methoxyamine in pyridine (10 mg/mL) (Sigma-Aldrich, St. Louis, MO, USA) was added to the dried pellet for 1 h at 70 °C. Subsequently, samples were derivatized with 100 μL of N-Methyl-N-(trimethylsilyl)-trifluoroacetamide, MSTFA, (Sigma-Aldrich, St. Louis, MO, USA) and incubated at room temperature for one hour. Finally, 50 μL of hexane was added to each sample and the solution was transferred to a vial for the GC-MS analysis.

### 2.6. Gas Chromatography-Mass Spectrometry Analysis

A 7890A gas chromatograph coupled with a 5975C Network mass spectrometer (Agilent Technologies, Santa Clara, CA, USA) equipped with a 30 m × 0.25 mm ID and fused silica capillary column, packaged with a 0.25 μM TG-5MS stationary phase (Thermo Fisher Scientific, Waltham, MA, USA), was exploited for metabolomic analysis. Samples were injected in splitless mode. The injector temperature was 250 °C whereas the transfer line was set at 280 °C. The carrier gas inside the column flowed at 1 mL/min rate. The programmed temperature was set as reported by Santoru and colleagues [37]: 60 °C for 3 min, then increased up to 140 °C at 7 °C/min and held at 140 °C for 4 min and finally raised to 300 °C at 5 °C/min and kept in isocratic mode at 300 °C for 1 min. Different sources were exploited for the identification of metabolites: NIST 08 (http://www.nist.gov/srd/mslist.cfm, accessed on 15 September 2022), GMD (http://gmd.mpimp/golm.mpg.de, accessed on 15 September 2022) mass spectra libraries and AMDIS software (Automated Mass Spectral Deconvolution and Identification System), freely available on www.amdis.net. Peak deconvolution, filtering, integration and normalization were performed using MassHunter Profinder Software from Agilent Technologies (La Jolla, CA, USA).

### 2.7. Glucose Uptake Assay

Glucose uptake was measured as previously described by Tronci et al. [33]. CRC cells were seeded in 96-well plates (1 × 10^5^ cells/mL) in a complete medium and incubated at 37 °C in order for attachment. After 24 h, cells were cultured in the presence or absence of glutamine for 48 h. Then, cells were rinsed with PBS and treated with 100 µL of a 50 µM solution of the fluorescent glucose analog 2-[N-(7-nitrobenz-2-oxa-1,3-diazol-4-yl) amino]-2-deoxy-D-glucose (2-NBDG, N13195; ThermoFisher, Waltham, MA, USA) in PBS in the presence or absence of glutamine. After an incubation of 30 min, the excess of 2-NBDG was removed and replaced with 100 µL PBS. Emitted fluorescence proportional to glucose uptake was read with a microplate reader (Infinite 200, Tecan, Salzburg, Austria). An excitation wavelength of 485 nm and an emission wavelength of 530 nm was exploited. All experiments were carried out in at least triplicate and repeated 3 times.

### 2.8. GLUT1 Protein Expression by Immunofluorescence

CRC-derived cells were cultured on previously sterilized slides and placed in square tissue culture dishes (quadriPERM^®^, Sarstedt AG & Co, Nümbrecht, Germany) for 24 h in a complete medium and in a humidified incubator. Thereafter, the medium was replaced and cells were grown for 48 h in the presence or absence of glutamine. After that, cells were fixed with 4% paraformaldehyde solution for 10 min at room temperature. As previously described [38], immunostaining was performed using rabbit polyclonal anti-GLUT1 (1:200; Abcam, CA, USA) antibody. The secondary antibody used was Alexa-conjugated (Alexa Fluor 488 or 594, Life Technologies) goat anti-rabbit IgG. To counterstain, nuclei 4′,6-diamidino-2-phenylindole (DAPI) was used. Images were obtained with an epifluorescence microscope (Olympus BX41) and charge-coupled device camera (Cohu), interfaced with the CytoVysion system (software 2.81 Applied Imaging, Pittsburg, PA, USA). Twenty randomly selected fields were acquired with a 20× objective for each cell line. The fluorescence intensity was determined by exploiting the Image J software (US National Institutes of Health, Bethesda, MD, USA).

### 2.9. 2-Deoxyglucose Treatment and MTT Viability Test

The inhibitory effect of 2-Deoxyglucose (2-DG) was evaluated in the presence or absence of glutamine using the MTT assay as follows. Briefly, cells were seeded at a density of 1 × 10^5^ cells/mL in a 96-well plate and incubated to allow for attachment. Then, cells were grown in a medium deprived of glutamine or not for 48 h. After that, the medium was washed off and cells were treated with 2-DG (2.5 mM for Caco-2, HCT116 and HT29, 5 mM for SW480) in the presence or absence of glutamine for another 48 h. After incubation, cells were rinsed with PBS and 50 µL of MTT solution (2 mg/mL in PBS) was added and left for 4 h at 37 °C. The resulting formazan crystals were solubilized in 100 μL of DMSO. The absorbance was measured using a TECAN microplate reader (Infinite 200, Tecan, Salzburg, Austria) at 570 nm [33]. Viability was calculated as % of control (cells grown in complete medium) for each cell line. All experiments were carried out in at least triplicate.

### 2.10. Determination of Intracellular Aminothyol Levels

The ratio between the reduced and oxidized form of glutathione (GSH/GSSG) was determined with high-performance liquid chromatography coupled with an electrochemical detector (HPLC-ECD), as previously described [39]. Cells were grown 4 days in complete medium in 6-well plates (density of seeding of 1 × 10^5^ cells/mL). After that, the medium was replaced, and cells were cultured in the presence or absence of glutamine for additional 48 h. Subsequently, cells were rinsed twice with PBS and extracted with 150 μL of 10% meta-phosphoric acid and 150 μL of 0.05% trifluoroacetic acid (Sigma-Aldrich, Milan, Italy) solution. The cell suspension was centrifugated and then the supernatant was injected into the HPLC system (Agilent 1260 infinity, Agilent Technologies, Palo Alto, Santa Clara, CA, USA) supplied with an electrochemical detector (DECADE II Antec, Leyden, The Netherlands) and an Agilent interface 35900E. A C-18 Phenomenex Luna with 5 μm particle size and 150 × 4.5 mm column was used. The mobile phase constituted 99% water with 0.05% TFA (*v/v*) and 1% MeOH, with a flow rate of 1 mL/min. The oxidizing potential of the electrochemical detector was set at 0.74 V. GSH and GSSH standards were injected before and after the samples run to allow for identification.

### 2.11. Statistical Analysis

GC-MS data obtained from MassHunter Profinder were organized into data matrices, where the columns represent the variables (area of the chromatographic peak), and the rows represent the samples. To minimize the effects of variable dilution of the samples, the final dataset was normalized to the total area. Each feature underwent univariate statistical analysis using GraphPad Prism software (version 7.01, GraphPad Software, Inc., San Diego, CA, USA) to identify statistically significant variables. The statistical significance of all performed experiments was assessed using the Student *t*-test, and a *p*-value of <0.05 was considered to be statistically significant.

## 3. Results

### 3.1. Glutamine Starvation Alters Growth Rate in Colorectal Cancer Cells

In order to investigate whether CRC cells were sensitive to glutamine deprivation, growth curves were evaluated. After seeding in a complete medium to allow for attachment, cells were grown in the presence or absence of glutamine, and viable cells were determined by an MTT assay every day. The results show that glutamine starvation significantly affected the growth rate in all studied CRC cell lines (Figure 1A–D). A significant decrease in viable cells was already highlighted on the first day of treatments in HCT116 and HT29 cells, whereas, in Caco-2 and SW480 cells, the statistical significance was reached after 48 h. After 3 days of starvation, the reduction in the growth in the glutamine deprivation condition was between approximately 40 and 50% in comparison to the respective control for all CRC cell lines (Table 2). Based on these results, glutamine represents an important amino acid for growth and its deprivation reduces the proliferation rate of colorectal cancer cell lines. 

### 3.2. Glutamine Starvation Reduces Proliferative Capacity in Colorectal Cancer Cells

The proliferation capacity during glutamine deprivation was evaluated with the colony-forming assay. Glutamine was essential for CRC cells to proliferate. Indeed, a significant reduction in the colony-forming ability was observed in all cell lines after starvation (Figure 2A–D). These results confirm that glutamine represented a pivotal amino acid, and its withdrawal altered the proliferation rate and growth capacity of colorectal cancer cells.

### 3.3. Glutamine Starvation Triggers Cell Death and G0/G1 Cell Cycle Arrest

Considering the decrease in the percentage of viable cells observed from the growth curves in the absence of glutamine, the influence of glutamine deprivation on apoptosis and necrosis was quantified using PI-Annexin V staining. In Caco-2 and HT29 cells, a significant decrease in the percentage of viable cells was observed upon glutamine deprivation (Figure 3A,C). In these cell lines, the necrotic percentage was higher in glutamine starvation than in the control condition, whereas only HT29 cells showed an increase in apoptotic cell death when cultured in the absence of glutamine (Figure 3C). In Caco-2, the mechanism of cell death by apoptosis was not involved, and, surprisingly, the apoptotic contribution was lower in the absence of glutamine compared to the control condition (Figure 3C). In SW480, the percentage of viable cells did not change when cells were deprived of glutamine; however, a significant increase in apoptotic percentage and a significant decrease in necrotic percentage were observed (Figure 3D). No effects of glutamine deficiency on the cell death of HCT116 cells were detected (Figure 3B). The percentage of viable, apoptotic and necrotic cells are detailed in Table 3. Because of these results, it was further investigated whether 48 h of glutamine starvation affected cell cycle progression by flow cytometry analysis. As shown in Figure 4A, 4B, 4C, 4D, glutamine withdrawal determined an accumulation of cells in the G0/G1 phase and a concomitant decrease in S and G2/M phases in all cell lines according to a significant decrease in the proliferation index (PI = (S + G2/M)/(G0/G1 + S + G2/M) *100) as reported in Table 4.

### 3.4. Metabolomic Alterations under Glutamine Deprivation

In order to investigate the metabolic alterations induced by glutamine deficiency, Caco-2, HCT116, HT29 and SW480 cells were grown in a complete or glutamine-deprived medium for 48 h. After this, hydrophilic intracellular metabolites were extracted and analyzed with GC-MS. Identified metabolites and statistical parameters are reported in Table 5. The metabolomic analysis showed a significant increase in D-glucose and D-galactose after glutamine starvation in all cell lines. Fructose and sorbose levels significantly increased in HT29 cells, whereas they decreased in Caco-2 cells. The glucose-6-phosphate level was higher in SW480 cells after starvation and ribose-5-phosphate was reduced in Caco-2, while mannose-6-phosphate decreased in Caco-2 and increased in SW480 cells. The GC-MS analysis showed a decrease in Krebs cycle intermediates. In particular, fumaric acid levels were lower in HCT116, HT29 and SW480 cells, while citric acid and malic acid concentrations were significantly lower in Caco-2 and HT29 cell lines. Lactic acid was reduced after glutamine starvation, especially in Caco-2, HCT116 and HT29 cells. The amino acidic pool was altered in glutamine withdrawal. In particular, glycine, phenylalanine, threonine and serine levels were significantly increased in all cell lines, whereas β-alanine, aspartic acid, glutamic acid and pyroglutamic acid amounts significantly decreased after starvation in all cell lines. Furthermore, aminomalonic acid and gamma-aminobutyric levels, tyrosine and valine were altered in Caco-2, HCT116 and SW480. Specifically, aminomalonic acid and gamma-aminobutyric acid decreased, whereas tyrosine and valine increased. Isoleucine, leucine and tryptophan levels increased after glutamine starvation in HCT116 and SW480 cells and creatinine was higher in Caco-2 and HCT116 cells, while alanine levels significantly decreased in Caco-2 and HT29 cell lines. The proline content was reduced only in HCT116 cells without glutamine. For polyols, a significant increase in myo-inositol was observed in all cell lines and the mannitol content was increased in Caco-2 cells and decreased in HCT116 cells, whereas the pantothenic acid content increased only in SW480 cells after starvation. The levels of some nucleosides or nitrogenous bases were reduced in glutamine withdrawal, especially adenosine monophosphate in Caco-2 and HT29, Uridine 5-monophosphate in HCT116 and 5′methylthioadenosine in Caco-2 cells. In addition, the glycerol-3-phosphate content increased after glutamine starvation in Caco-2, HCT116 and SW480 cells, while cholesterol levels increased in Caco-2 cells. Finally, the levels of beta-glycerophosphoric acid, myo-Inositol 1-phosphate, niacinamide and taurine showed no significant changes in glutamine-deprived conditions.

### 3.5. Glutamine Deprivation Altered Glucose Uptake and GLUT1 Expression

Considering the increase in sugar levels observed by the metabolomic analysis following glutamine deprivation, it was hypothesized that their increase could be due to an enhanced glucose uptake during starvation. Therefore, the glucose uptake was evaluated using a fluorescent analog, 2-NBDG. Furthermore, the expression GLUT1 glucose transporter was evaluated. A significant increase in 2-NBDG uptake was observed in Caco-2 and SW480 cells (Figure 5A,D), whereas only an increasing trend was detected in HCT116 and HT29 cell lines after 48 h of glutamine deprivation (Figure 5B,C). 

Moreover, the GLUT1 was significantly over-expressed in Caco-2, HCT116 and SW480 cells after 48 h of glutamine starvation (Figure 6Ai,Bi,Di). Only HT29 cells showed a remarkable decrease in GLUT1 expression after glutamine starvation (Figure 6Ci). These results suggest that glutamine deprivation can alter glucose transport across the plasma membranes by inducing GLUT1 expression.

### 3.6. The Combined Treatment with Glutamine Deprivation and 2-Deoxy-D-Glucose Affected CRC Survival more than Glutamine Deprivation

Given the increased glucose avidity of cancer cells during glutamine starvation, it has been speculated that combining the starvation with a glycolysis inhibitor, 2-DG, could affect cancer cells’ survival and proliferation. The dose of 2-DG used is different for each cell line. Therefore, it was considered useful to use a concentration able to induce a decrease in vitality between 30 and 50% compared to the control. For this purpose, the Caco-2, HCT116 and HT29 cells were treated with 2.5 mM 2-DG and SW480 cells with 5 mM 2-DG in the presence or absence of glutamine for 48 h. Cell viability was assessed using the MTT assay. Figure 7B–7D shows that the combination of the two treatments (2-DG and glutamine starvation) results in a significantly lower percentage of viable cells in comparison with both glutamine starvation and 2-DG treatment alone in HCT116, HT29 and SW480 cells. Only in Caco-2 cells does the combined treatment not show significant synergistic effects compared to glutamine deprivation alone. The result suggests that targeting glucose and glutamine metabolism simultaneously could represent a more effective strategy for suppressing the survival of colorectal cancer cells.

### 3.7. Glutamine-Deprivation-Induced Reduction in Antioxidant Defenses in CRC Cells

Glutamate synthesized from glutamine is one of the components of glutathione, an important cellular antioxidant [20]. Consequently, in order to evaluate the alterations in redox balance induced by glutamine starvation, the reduced and oxidized glutathione forms (GSH and GSSG, respectively) were evaluated by HPLC analysis. In all cell lines examined, a significant decrease in GSH/GSSG ratios was observed after 48 h of glutamine starvation compared with the respective control grown in a complete medium (Figure 8A–D). This observation highlights the role of glutamine in maintaining adequate antioxidant defenses.

## 4. Discussion

Glutamine fuels energetic and biosynthetic processes and ensures redox homeostasis in cells, playing a pivotal role, especially in high-proliferating cells [40,41]. In particular, cancer cells rewire glutamine metabolism and become addicted to this amino acid, such as CRC cells [16,17,42,43,44,45]. In this context, glutamine metabolism represents an intriguing target for investigation. Exploring and understanding how glutamine is involved in cancer cell survival, proliferation and redox homeostasis is essential for exploiting aberrant tumorigenic metabolism in the therapeutic field. In the present study, proliferation, survival, antioxidant species and the metabolic profile were investigated in four CRC cell lines. The results confirm that CRC cells were sensitive to glutamine starvation. Deprivation resulted in a marked antiproliferative effect, with a decrease in growth rate ranging from 40 to 50 percent in all cell lines. It has been demonstrated that glutamine deprivation can induce cell death or block cell proliferation depending on the cell line [40,45,46,47,48]. In this study, changes in the apoptosis and cell cycle of colon cancer cells grown in the absence of glutamine were examined. The results demonstrate that the depletion of glutamine inhibited cell proliferation in the colon cancer cells via apoptosis or necrosis, and induced cell cycle G0/G1 arrest. These observations are in agreement with previous data [40] and indicate that the anti-proliferative effects exerted by glutamine deprivation can be attributed to the induction of cell cycle arrest and cell death. Several mechanisms come into play in the activation of the apoptotic pathway following glutamine deprivation [49,50,51,52]. In our cells, cell death could be induced by the reduction in the reduced form of GSH, as observed through HPLC analysis. Moreover, in HCT116 cells, glutamine starvation does not induce cell death, but it is well-known that the absence of glutamine causes a distinct response in different cell types [53]. Instead, HCT116 cells reduced their proliferation rate by entering a quiescent state, as demonstrated by the augmentation of cells in the G0/G1 phase as demonstrated by others in different cell lines [54,55,56]. The decrease in the cell population in the synthetic and mitotic phases is consistent with the importance of glutamine in the overcoming of the G1 phase and the synthesis of nucleotides [54,57,58]. Overall, glutamine deprivation reduces proliferating cells, and, at the same time, the formation of cancer cell colonies is decreased in all cell lines. The results suggest an effort to overcome starvation by down-regulating cell cycle progression and avoiding cell death. Moreover, data demonstrated that cancer cells can adapt to overcome glutamine withdrawal through the adjustment of specific metabolic pathways. Moreover, the withdrawal of glutamine can induce a decrease in the levels of TCA cycle intermediates, as revealed by the GC-MS analysis. In particular, the level of citrate, fumarate and malate was significantly decreased in almost all cell lines after 48 h of glutamine deprivation. Moreover, the GC-MS analysis also displayed higher glucose levels in glutamine-deprived cells. To explain this increase, the glucose uptake was evaluated, as well as the expression of GLUT1, one of the most important glucose transporters. A strongly enhanced glucose uptake was observed in Caco-2 and SW480 cells, whereas, in HCT116 and HT29 cells, only an increasing trend was noted. Furthermore, the expression of GLUT1 was increased after glutamine starvation in Caco-2, HCT116 and SW480 cells. Surprisingly, GLUT1 expression was reduced in HT29 cells, despite increased intracellular glucose levels. This controversial result could be explained by the overexpression of GLUT3, another glucose transporter in the HT29 cell line as already observed by Kuo et al. [59]. Furthermore, it has been shown that cancer cell growth could be reduced by inhibiting both glucose transporters and glutamine metabolism, and this represents a successful strategy for cancer therapy [60,61]. Despite the higher glucose uptake, lactate levels were decreased after glutamine withdrawal. It is probable that the glycolytic rate, and, consequently, the energy production, were downregulated to exploit glucose for biosynthetic purposes. Moreover, glucose can be used to produce NAPDH through the pentose phosphate pathway [62], regenerate GSH from GSSG and counteract oxidative stress, as already proven by Cetinbas and colleagues [63]. In the pancreatic β-cell line during glutamine starvation, glucose’s carbons were used to synthesize glutamate, usually produced from glutamine [64,65]. As expected, the metabolomics analysis showed that the glutamate level and its cyclized form, pyroglutamic acid, were significantly decreased in glutamine deprivation conditions in all cell lines. Indeed, glutamate is directly produced from glutamine via deamination by GLS [66]. The threonine, tryptophan, tyrosine and valine levels were significantly higher in almost all studied cell lines after 48 h of glutamine starvation. Supposedly, glutamine absence triggered the uptake of exogenous amino acids, as observed by Chen and colleagues [67]. In addition, glycine and serine intracellular concentrations were markedly enhanced in all cell lines after glutamine withdrawal. This is in accordance with Tanaka and colleagues, who found a raised level of serine in glutamine-deprived glioblastoma cells [68]. Serine and glycine amino acids participate in several processes, including amino acids, purines, antioxidants synthesis and the folate cycle through one-carbon metabolism [69,70]. The enhanced levels of these amino acids could support cells in maintaining redox homeostasis in glutamine deprivation conditions. In addition, in our colon cancer cells, the level of beta-alanine is altered following glutamine deprivation; specifically, its concentration is significantly decreased in starved cells. The beta-alanine level was found to be significantly upregulated in human CRC tissues [71]. Furthermore, Hutschenreuther and colleagues correlated beta-alanine acid with lactate concentration and with glycolytic activity [72]. This is in agreement with the decrease in lactate production in all colon cancer cell lines. In addition, glutamine is mainly converted into alanine and lactate and then carried out in the extracellular space [73,74]. In glutamine absence, pyruvate or alanine refuels the Krebs cycle through an anaplerotic flux to compensate for glutamate deficiency. This process has been suggested as a resistant mechanism for pharmacological glutaminase inhibition in cancer cells [75,76,77]. These observations could underlie the pronounced decrease in alanine and lactate seen in our glutamine-deprived cells. Thereby, this mechanism could provide energetic and biosynthetic substrates, without, however, overcoming the inhibition caused by glutamine withdrawal. Moreover, the metabolomic analysis highlighted a significant decrease in myo-inositol levels in cells deprived of glutamine. Myo-inositol plays a crucial role in cell survival and proliferation [78]. This should be consistent with the lower proliferation capacity observed. As noted from metabolomic analysis and glucose uptake assay, cells deprived of glutamine uptook more glucose. Thus, it is possible to hypothesize that, during starvation, tumor cells put in place this strategy to overcome glutamine depletion and survive. As expected, glutamine deprivation combined with glucose metabolism inhibition with 2-DG exerted a greater cytotoxic effect than the treatments considered individually in all studied CRC cell lines. As reported by Le and colleagues, glutamine compensates for energetic and biosynthetic purposes when the tumor cells are glucose-deficient [79]. Conversely, glutamine depletion led to an increased glucose utilization [65]. At the same time, blocking both glucose and glutamine utilization markedly affected the cell viability [60]. Therefore, an imbalance of glutaminolysis and glucose metabolism could cause metabolic distress [80]. It is well known that glutamine is indirectly involved in maintaining cellular redox homeostasis. Indeed, glutamine represents the primary precursor of glutamate, which, in turn, is a component of glutathione, one of the major cellular antioxidant species [81]. The metabolomic analysis showed a significant decrease in glutamate content (and its cyclized form, pyroglutamic acid) after 48 h of glutamine withdrawal in all CRC cell lines. Moreover, a significant decrease in antioxidant species, as shown by the change in the ratio between GSH/GSSG, indicates an increased oxidative condition. The oxidative stress damage could be responsible for the decreased cell proliferation and apoptosis/necrosis induction, although the underlying mechanism is still unclear [82]. Overall, although the CRC cell lines studied harbor the common genetic mutations (APC, KRAS, PIK3CA and TP53 genes) associated with the different tumor stages [83], in our hands, glutamine starvation did not determine relevant differences in survival, proliferation, oxidative homeostasis and metabolism. These in vitro results suggest that colon cancer cells, independent of genetic and phenotypic features, implement a similar strategy to overcome the withdrawal of this nutrient. Nevertheless, in vivo studies are needed to confirm this hypothesis.

## 5. Conclusions

Glutamine represents a pivotal metabolite for tumor viability and proliferation by fulfilling the metabolic and biosynthetic request and by regulating the redox homeostasis counteracting the increase in oxidative species during rapid proliferation. Indeed, our results suggest that glutamine depletion deeply altered energy metabolism, the amino acid content and antioxidant defenses, and induced similar metabolic alterations in colon cancer cells. These metabolic perturbations could be responsible for the decreased cell proliferation and apoptosis/necrosis independently by genetic and phenotypic features of colon cancer. This suggests that tumor cells try to overcome starvation by putting in place a common metabolic rewiring. This offers a strong rationale for targeting glutamine and glucose metabolism and regulating the redox status to achieve significant therapeutic benefits in the clinic.

## Figures and Tables

**Figure 1 antioxidants-12-00683-f001:**
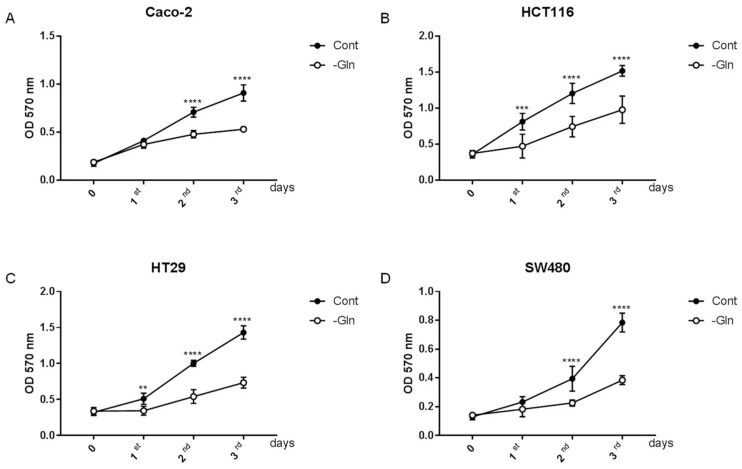
Growth curves of colorectal cancer cell lines ((**A**) Caco-2, (**B**) HCT116, (**C**) HT29, (**D**) SW480) under glutamine starvation for 3 days. Cell viability was assessed by MTT assay at the indicated time points in the presence (●) or absence (o) of glutamine in the medium. Representative growth curves of three independent experiments are shown. Data are presented as mean ± standard deviation (SD) (*n* = 3, ** *p* < 0.005, *** *p* < 0.0005, **** *p* < 0.0001 compared to Cont).

**Figure 2 antioxidants-12-00683-f002:**
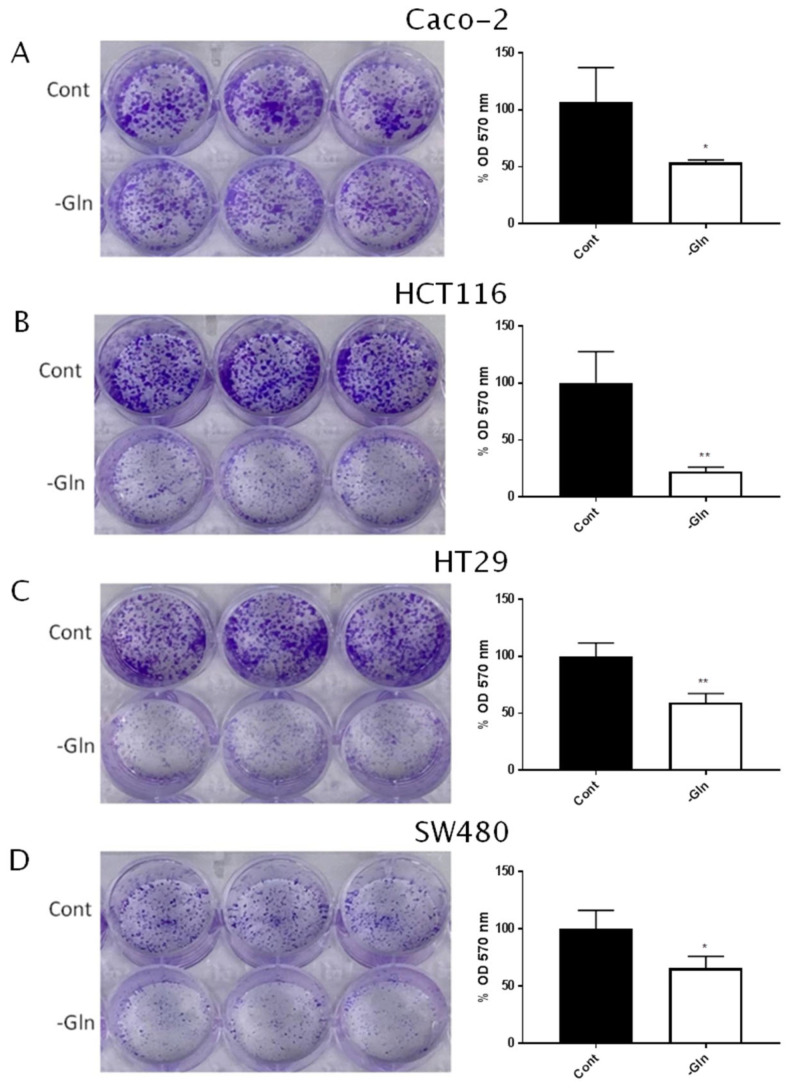
Glutamine deprivation reduced the colony-forming ability of colorectal cancer cells ((**A**) Caco-2, (**B**) HCT116, (**C**) HT29, (**D**) SW480). Cells were seeded in 6-well plates (500–1000 cells/well) and left to grow for 48h in a complete medium. Cells were then shifted to glutamine-deprived medium (-Gln) or to complete medium (Cont) and were fixed with crystal violet after 6–10 days. Optical density (OD 570 nm) of solubilized crystal violet was used to evaluate cell colony formed. Data are presented in histograms as a percentage of control. The experiment was performed in triplicate and represented mean ± SD (*n* = 3, * *p* < 0.05, ** *p* < 0.005).

**Figure 3 antioxidants-12-00683-f003:**
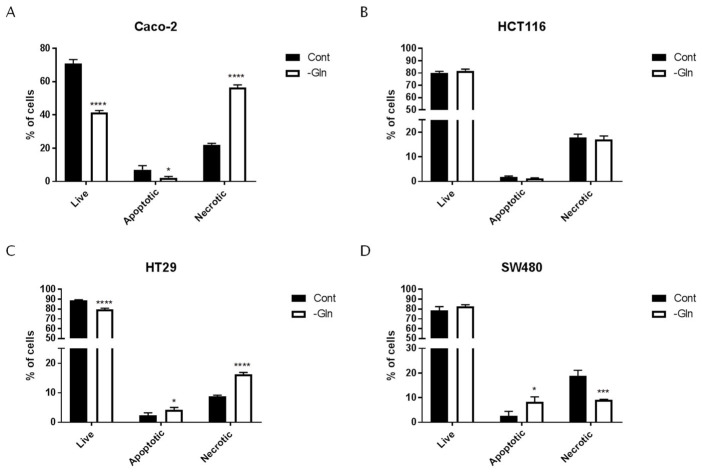
Analysis of cell death in colorectal cancer cells ((**A**) Caco-2, (**B**) HCT116, (**C**) HT29, (**D**) SW480). Cell death was evaluated in colorectal cancer cells in the presence (Cont) or absence (-Gln) of glutamine for 48 h using flow cytometry combined with annexin V and propidium iodide staining. Data are presented as cell percentage; the experiment was performed in triplicate and represents mean ± SD (*n* = 3, * *p* < 0.05, *** *p* < 0.0005, **** *p* < 0.0001 compared with Cont).

**Figure 4 antioxidants-12-00683-f004:**
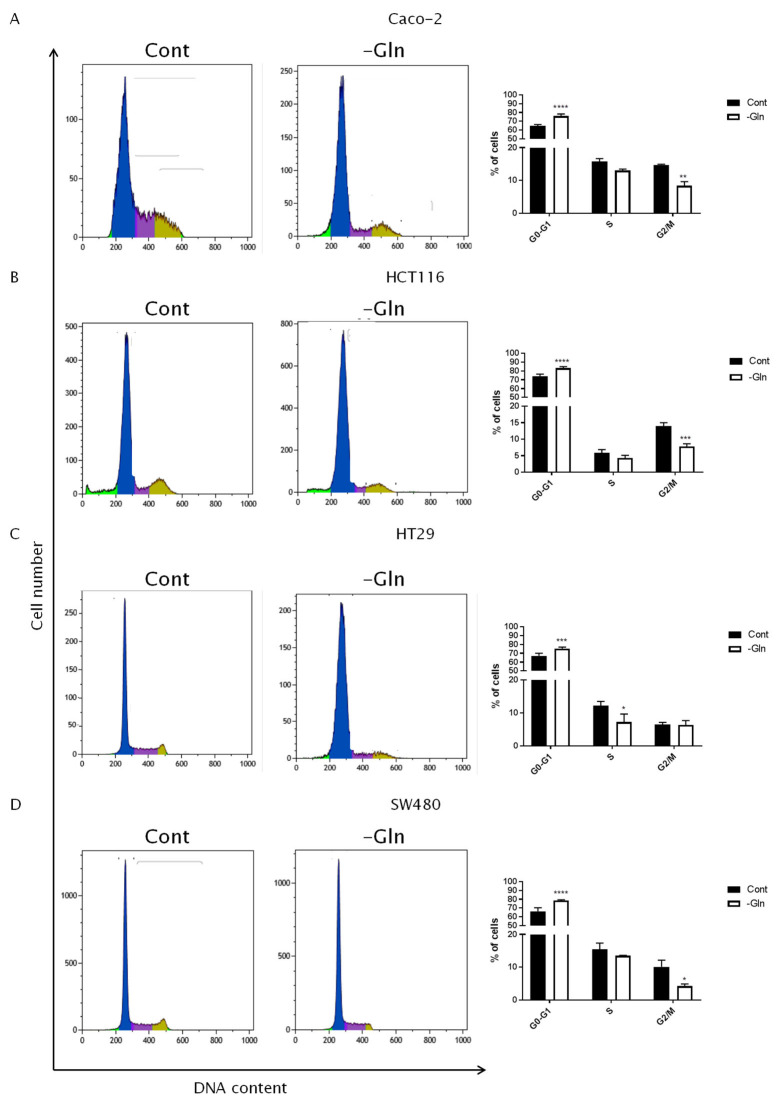
Flow cytometry analysis for cell cycle distribution of CRC cells in the presence or absence of glutamine. Glutamine-starvation-induced cell cycle arrest in colorectal cancer cells ((**A**) Caco-2, (**B**) HCT116, (**C**) HT29, (**D**) SW480). The result of one representative assay from three similar independent experiments is shown. *x*- and *y*-axes denote cell number and DNA content, respectively. The histograms show the statistical representation of the percentage of the cell population at the different phases of the cell cycle distribution; the experiment was performed in triplicate and represents mean ± SD (*n* = 3, * *p* < 0.05, ** *p* < 0.005, *** *p* < 0.0005, **** *p* < 0.0001 compared to Cont).

**Figure 5 antioxidants-12-00683-f005:**
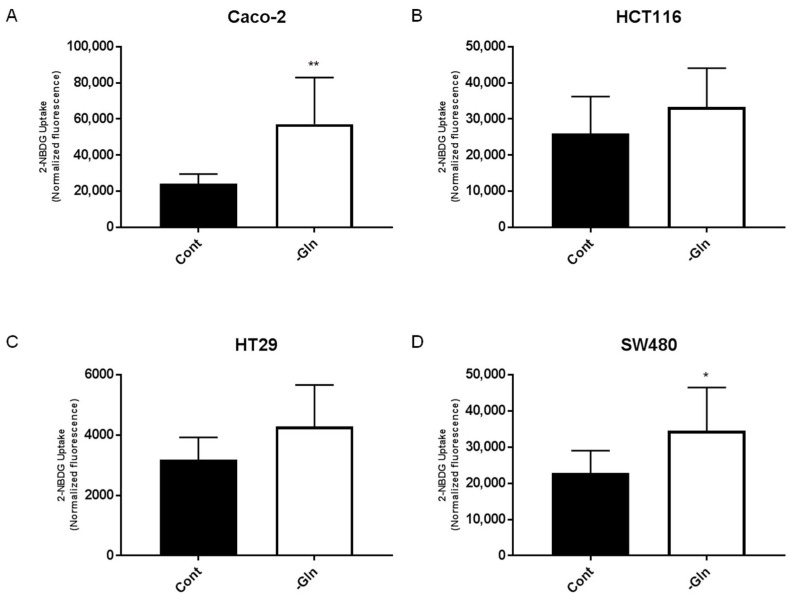
Colorectal cancer cell lines exhibited changes in glucose uptake under glutamine deprivation ((**A**) Caco-2, (**B**) HCT116, (**C**) HT29, (**D**) SW480). Glucose uptake was measured through the quantification of the fluorescence emitted by the glucose analog 2-NBDG in cancer cells cultured in the presence or the absence of glutamine 4 mM for 48h. Data are presented as mean ± SD (*n* = 3). Statistical analysis was performed by Student t-Test. Results were considered significant when * *p* < 0.05, ** *p* < 0.05.

**Figure 6 antioxidants-12-00683-f006:**
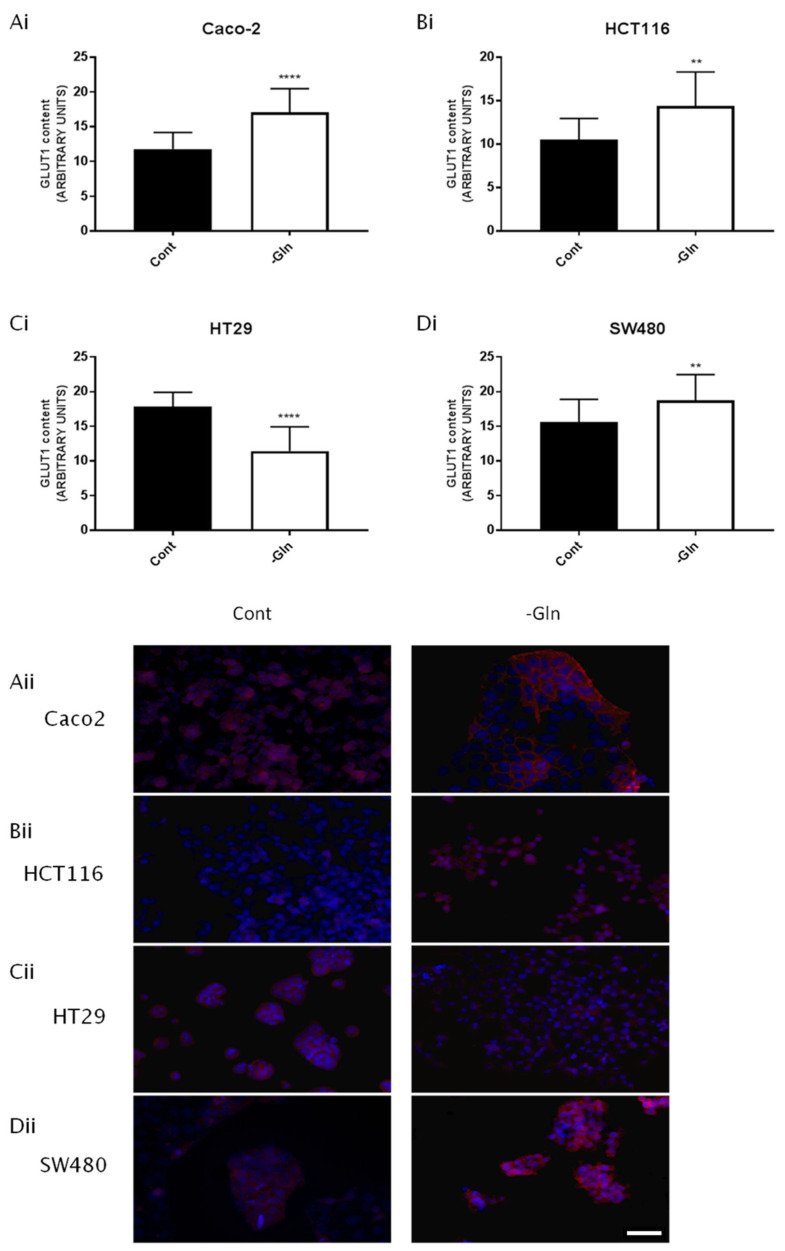
Glutamine starvation altered the expression of glucose transporter GLUT1. Quantification of immune fluorescent pattern in the colorectal cancer cell lines after 48h of glutamine deprivation (-Gln) and relative controls (Cont). (**Ai**) Caco-2, (**Bi**) HCT116, (**Ci**) HT29, (**Di**) SW480. Data are shown as percentage of control. Statistical analysis was performed by Student *t*-test. The experiment was performed in triplicate and represented mean ± SD (*n* = 3, ** *p* < 0.005, **** *p* < 0.0001 compared to Cont). (**Aii**) Caco-2, (**Bii**) HCT116, (**Cii**) HT29, (**Dii**) SW480 microphotographs (Scale bar = 10 μm, magnification = 20×) of colorectal cancer cells with GLUT1 stained in red and nuclei stained with DAPI in blue.

**Figure 7 antioxidants-12-00683-f007:**
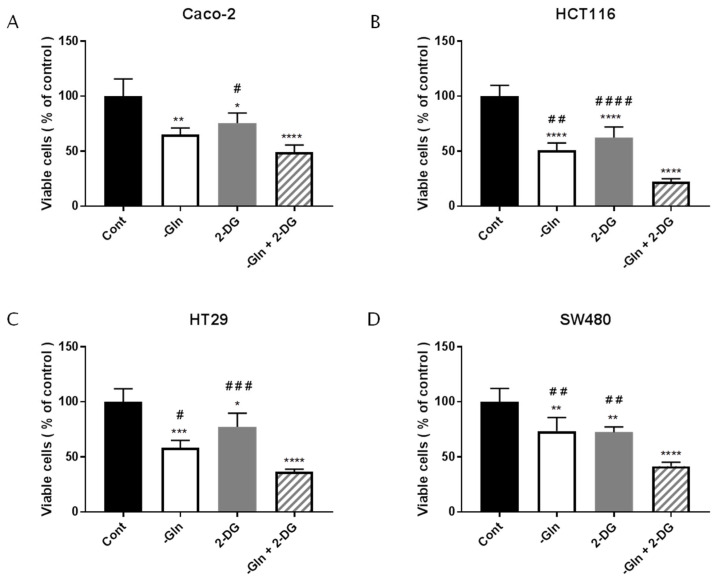
Combined treatment with glutamine deprivation and 2-DG induced stronger cytotoxic effects in colorectal cancer cells ((**A**) Caco-2, (**B**) HCT116, (**C**) HT29, (**D**) SW480). Cells were treated with 2-DG (2.5 mM Caco-2, HCT116 and HT29, 5 mM SW480 cells) in the presence or absence of Gln (Cont) for 48h. Viable cells were evaluated with MTT assay. Data are presented as mean ± SD (*n* = 3, * *p* < 0.05, ** *p* < 0.005, *** *p* < 0.0005, **** *p* < 0.0001 compared with Cont; # *p* < 0.05, ## *p* < 0.005, ### *p* < 0.0005, #### *p* < 0.0001 compared with combined treatment). One-way ANOVA statistical analysis was performed.

**Figure 8 antioxidants-12-00683-f008:**
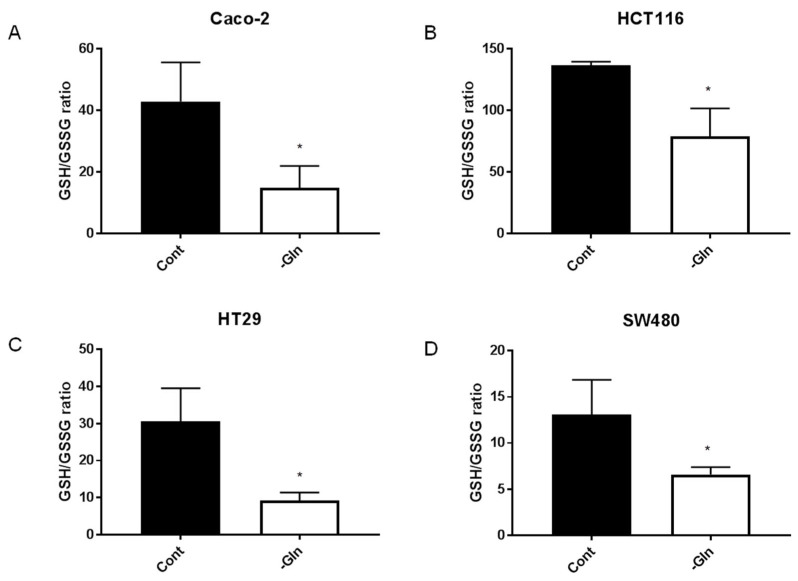
Glutamine starvation significantly decreased intracellular antioxidant defenses in (**A**) Caco-2, (**B**) HCT116, (**C**) HT29, (**D**) SW480. GSH and GSSG levels were measured by HPLC in colorectal cancer cells after 48h of glutamine deprivation condition. Experiments were performed in triplicate and data were expressed as mean ± SD (*n* = 3, * *p* < 0.05).

**Table 1 antioxidants-12-00683-t001:** Colorectal cancer cell lines, genetic background and classification.

Cell line	Caco-2	HCT116	HT29	SW480	References
APC	c.4099C>T	Wild type	c.2557G>T	c.4012C>T	[27,28]
BRAF	Wild type	Wild type	c.1799T>A	Wild type	ATCC [27,29]
CTNNB1	c.734G>C	c.133_ 135delTCT	Wild type	Wild type	[27,30]
KRAS	Wild type	c.38G>A	Wild type	c.35G>T	[27,28]
PIK3CA	Wild type	c.3140A>G	c.1345C>A	Wild type	[27]
SMAD4	c.1051G>C	Wild type	c.931C>T	Wild type	[27,28]
TP53	c.610G>T	Wild type	c.818G>A	c.818G>A	[27,31]
Histotype	Colorectal carcinoma	Colorectal carcinoma	Colorectal adenocarcinoma	Colorectal adenocarcinoma	[29]
Duke’s stage	/	D	C	B	[29]

**Table 2 antioxidants-12-00683-t002:** Cell viability after 3 days of glutamine starvation in colorectal cancer cell lines.

Cell Lines	Cell Viability in -Gln Condition (% of Controls)	±SD
Caco-2	55.02	5.04
HCT116	62.01	3.25
HT29	48.83	7.07
SW480	49.34	0.57

The reported percentages represent the mean of at least three independent experiments ± SD.

**Table 3 antioxidants-12-00683-t003:** The influence of glutamine deprivation on colorectal cancer cell death.

Cell Lines		Live		Apoptotic		Necrotic	
		% of Cells ±SD	*p*-Value (vs. Cont)	% of Cells ±SD	*p*-Value (vs. Cont)	% of Cells ±SD	*p*-Value (vs. Cont)
Caco-2	Cont	70.93 ± 2.39		7.05 ± 2.48		22.02 ± 0.86	
	-Gln	41.36 ± 1.27	<0.0001	2.20 ± 0.86	0.0025	56.44 ± 1.67	<0.0001
HCT116	Cont	80.24 ± 1.15		1.84 ± 0.33		17.93 ± 1.28	
	-Gln	81.73 ± 1.46	*ns*	1.24 ± 0.21	*ns*	17.02 ± 1.48	*ns*
HT29	Cont	88.92 ± 0.49		2.32 ± 0.90		8.77 ± 0.40	
	-Gln	82.36 ± 0.56	<0.0001	4.16 ± 0.88	0.0395	16.24 ± 0.63	<0.0001
SW480	Cont	78.43 ± 4.00		2.66 ± 1.80		18.91 ± 2.22	
	-Gln	82.48 ± 2.60	*ns*	8.33 ± 2.83	0.0311	9.19 ± 0.23	0.0030

Cell death was evaluated in colorectal cancer cells in the presence (Cont) or absence (-Gln) of glutamine for 48 h using flow cytometry combined with annexin V and propidium iodide staining. Data were expressed as cell percentages; the experiment was performed in triplicate and represents mean ± SD; *ns*: not significant.

**Table 4 antioxidants-12-00683-t004:** Glutamine-starvation-induced cell cycle arrest.

Cell Line		G0/G1 Phase		S Phase		G2/M Phase		PI	
		% of Cells ±SD	*p*-Value (vs. Cont)	% of Cells ±SD	*p*-Value (vs. Cont)	% of Cells ±SD	*p*-Value (vs. Cont)	% of Cells ±SD	*p*-Value (vs. Cont)
Caco-2	Cont	64.71 ± 1.57		15.85 ± 0.84		14,76 ± 0.20		30.61 ± 1.04	
	-Gln	76.12 ± 2.18	<0.0001	13.06 ± 0.45	*ns*	8.41 ± 1.25	0.0017	21.46 ± 1.59	0.0059
HCT116	Cont	73.77 ± 2.71		5.79 ± 1.03		13.98 ± 1.01		19.77 ± 2.03	
	-Gln	83.28 ± 1.75	<0.0001	4.28 ± 0.79	*ns*	7.81 ± 0.79	0.0009	12.09 ± 0.31	0.0029
HT29	Cont	67.10 ± 3.02		12.16 ± 1.35		6.51 ± 0.62		18.66 ± 1.95	
	-Gln	75.45 ± 1.64	0.0005	7.22 ± 2.42	0.0243	6.31 ± 1.40	*ns*	13.54 ± 1.67	0.0258
SW480	Cont	66.21 ± 4.11		15.49 ± 1.87		10.12 ± 2.08		25.61 ± 3.62	
	-Gln	78.61 ± 0.86	<0.0001	13.50 ± 0.14	*ns*	4.26 ± 0.65	0.0143	17.77 ± 0.79	0.0215

Colorectal cancer cell lines were grown in glutamine-deprived medium for 48 h, stained with propidium iodide and subjected to flow cytometric analysis. The proliferating index was calculated as PI = (S + G2/M)/(G0/G1 + S + G2/M) * 100. Experiments were performed in triplicate and data are expressed as cell percentages and represent mean ± SD; *ns*: not significant.

**Table 5 antioxidants-12-00683-t005:** Statistical parameters of metabolites detected with GC-MS analysis after 48 h of glutamine deprivation.

Metabolites	Caco-2		HCT116		HT29		SW480	
	Fold Change -Gln/Cont	*p*-Value	Fold Change -Gln/Cont	*p*-Value	Fold Change -Gln/Cont	*p*-Value	Fold Change -Gln/Cont	*p*-Value
Adenosine monophosphate	0.60	0.0135	0.40	*ns*	0.48	0.0017	0.82	*ns*
Aminomalonc acid	1.62	0.0003	4.04	0.0012	-	-	3.96	0.0154
Beta-Alanine	0.36	0.0007	0.51	0.0054	0.43	<0.0001	0.51	0.0016
Beta-Glycerophosphoric acid	1.02	*ns*	1.23	*ns*	-	-	-	-
Cholesterol	1.56	0.0169	0.98	*ns*	0.56	*ns*	0.87	*ns*
Citric acid	0.36	<0.0001	0.85	*ns*	0.69	0.0029	0.96	*ns*
Creatinine	1.67	0.0059	1.40	0.0342	-	-	-	-
D-Fructose	0.80	0.0010	0.93	*ns*	1.75	0.0062	-	-
D-Galactose	1.53	0.0004	2.57	0.0021	1.50	0.0065	3.82	0.0042
D-Glucose	1.66	0.0011	2.47	0.0039	1.51	0.0049	4.11	0.0030
D-Glucose 6-phosphate	1.42	*ns*	1.31	*ns*	0.92	*ns*	1.42	0.0036
D-Ribose 5-phosphate	0.29	0.0010	0.77	*ns*	0.49	*ns*	1.56	*ns*
Fumaric acid	0.05	*ns*	0.69	0.0022	0.14	<0.0001	0.53	0.0381
Gamma-Aminobutyric acid	0.19	0.0025	0.36	0.0210	1.08	*ns*	0.28	0.0199
Glycerol 3-phosphate	1.37	0.0256	1.01	*ns*	1.58	<0.0001	1.95	0.0008
Glycine	3.56	<0.0001	3.87	<0.0001	8.44	<0.0001	3.68	0.0002
L-Alanine	0.13	0.0070	0.46	*ns*	0.20	0.0431	0.71	*ns*
L-Aspartic acid	0.27	0.0039	0.29	0.0050	-	-	0.16	0.0091
L-Glutamic acid	0.13	0.0021	0.72	0.0109	0.31	0.0012	0.26	0.0270
L-Isoluecine	1.08	*ns*	2.27	0.0068	-	-	2.20	0.0193
L-Lactic acid	0.67	0.0039	0.30	<0.0001	0.54	0.0001	0.92	*ns*
L-Leucine	0.29	*ns*	2.71	0.0011	-	-	2.50	0.0024
L-Phenylalanine	2.00	0.0064	4.08	<0.0001	-	-	3.53	0.0016
L-Proline	0.97	*ns*	0.41	0.0068	-	-	0.86	*ns*
L-Sorbose	0.83	0.0114	1.06	*ns*	1.93	0.0004	-	-
L-Threonine	2.31	<0.0001	5.07	<0.0001	2.63	0.0121	2.99	<0.0001
L-Tryptophan	1.34	*ns*	2.14	<0.0001	-	-	1.99	0.0030
L-Tyrosine	2.69	0.0007	3.15	0.0006	2.42	*ns*	3.16	0.0014
L-Valine	1.17	0.0242	2.92	0.0093	1.63	*ns*	1.74	0.0244
Malic acid	0.13	<0.0001	0.79	*ns*	0.14	<0.0001	0.51	*ns*
Mannitol	1.99	0.0022	0.54	0.0006	-	-	1.60	0.0095
Mannose 6-phosphate	0.48	0.001	1.38	*ns*	0.81	*ns*	1.36	0.0180
Myo-Inositol	0.78	0.0093	0.65	0.0010	0.63	0.0001	0.38	<0.0001
Myo-Inositol 1-phosphate	1.03	*ns*	0.86	*ns*	1.07	*ns*	0.92	*ns*
Niacinamide	-	-	0.86	*ns*	-	-	0.98	*ns*
Pantothenic acid	0.92	*ns*	0.90	*ns*	0.92	*ns*	1.42	0.0112
Pyroglutamic acid	0.10	0.0022	0.12	0.0333	0.13	<0.0001	0.16	0.0011
Serine	7.14	<0.0001	7.94	<0.0001	13.71	0.03052	2.46	0.0003
Taurine	2.21	*ns*	0.90	*ns*	-	-	1.10	*ns*
Uridine 5-monophosphate	1.09	*ns*	0.42	0.0001	0.40	*ns*	-	-
5′Methylthioadenosine	0.32	0.0498	1.07	*ns*	0.94	*ns*	0.88	*ns*

Fold changes were calculated as the ratio of glutamine-deprived samples and control. Values above 1 correspond to up-regulated concentrations, whereas values below 1 correspond to down-regulated concentrations; *ns*: not significant. The dash indicates that the metabolite was not identified in that sample.

## Data Availability

The data presented in this study are shown in this paper.

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
