# Peer review of "Glutamine Starvation Affects Cell Cycle, Oxidative Homeostasis and Metabolism in Colorectal Cancer Cells"

_antioxidants, 2023, doi:10.3390/antiox12030683_

Round 1
Reviewer 1 Report
In this manuscript, the authors investigated the effect of glutamine starvation on the cell cycle and metabolism in colorectal cancer cells. Glutamine starvation inhibited cell growth and colony formation by affecting the cell cycle and decreasing intracellular antioxidant defenses. In addition, the authors showed that glutamine depletion induced intracellular glucose levels, and this phenomenon was caused in part by altered expression of glucose transporter 1. They also showed that glutamine deprivation and 2-deoxyglucose-induced cytotoxic effects in colorectal cancer cells. The methods used here are pragmatically accurate, and the manuscript contains original information on the effect of glutamine starvation on the metabolism of colorectal cancer cells. The following minor points need to be addressed.
1. In Table 4, the authors should show the ratio of metabolites (glutamine deprivation vs. control).
2. Please use larger fonts in Figure 4 to improve image clarity for the reviewers and readers.
3. In Figure 4, which graph shows the cell cycle induced by glutamine deprivation?
4. In Figure 6, what is the difference between the left and right panels? Also, the authors should provide scale bars in Figure 6.
5. Line 321: deprived cellsAbout the →deprived cells. About the (?)
Reviewer 2 Report
The manuscript “Glutamine starvation affects cell cycle, oxidative homeostasis 2 and metabolism in colorectal cancer cells” by Spada et al. is a contribution to the already well-established role of L-glutamine in cancer metabolism. The paper describes the influence of of glutamine, particularly in different biological processes in CRC cells such as proliferation rate, apoptosis, metabolomics etc. where it compares the results with or without glutamine. The paper is well structured and the experiments are done with precision. The main question of this reviewer is whether it is really within the scope of this journal since the connection with antioxidants or oxidative stress is week, and definitely not in the focus of this research. Moreover, there is no clear statement what is the novelty of this research. If there is substantial novelty it should be clearly emphasized. The mechanistic explanations or at least theories are missing. Therefore, although the experimental part is performed lege artis I am uncertain that this paper is suitable for Antioxidants.
Major:
1. Please state the crucial novelty of your work because it is not clearly stated. The role of glutamine in cancer cells is known to broad community. You should make clear to the reader what the novelty of your research is.
2. Please describe why you used these 4 types of CRC cells (they are different in origin and differentiation status). Explain the differences that occurred in your bioassays among the tested cell lines. This was not mentioned in the discussion. Do you have any explanations for the differences you observed? Please discuss this in the appropriate section.
3. What happens to normal cells in the body when you deprive them of glutamine if you use this procedure as a therapeutic?
4. Why wait 4 days in many experiments (growth rate, colony forming etc) to deprive the cells glutamine. It was not explained.
5. Some, at least theoretical explanations of why and how your observations concerning the role of glutamine could be explained is missing. The conclusion is quite unenthusiastically written.
Minor:
The paper is written quite understandably but crosschecking English language would be beneficial. Some sentences are weird. For example:
Line 268- Any effects of…..-the sentence is unclear, possibly English is not correct
Line 271- replace next with further
Line 320 - Proline content etc…this sentence and the next About the polyols…there is clearly something wrong with these sentences
Reviewer 3 Report
It is a well written study, however the authors should discuss the possibility of the observed differences on glucose uptake and other parameters to be due to the different mutation status of the cell lined used.
Round 2
Reviewer 2 Report
The manuscript can be accepted. The authors answered adequately to my comments.